# Increasing the Adhesion of Bitumen to the Surface of Mineral Fillers through Modification with a Recycled Polymer and Surfactant Obtained from Oil Refining Waste

**DOI:** 10.3390/polym16050714

**Published:** 2024-03-05

**Authors:** Antonina Dyuryagina, Yuliya Byzova, Kirill Ostrovnoy, Alexandr Demyanenko, Aida Lutsenko, Tatyana Shirina

**Affiliations:** Department of Chemistry and Chemical Technology, Manash Kozybayev North Kazakhstan University, Petropavlovsk 150000, Kazakhstan; adyuryagina@inbox.ru (A.D.); kostrovnoy@mail.ru (K.O.); demianenkoav@mail.ru (A.D.); ll-a.13@mail.ru (A.L.); tshirina@internet.ru (T.S.)

**Keywords:** optimization of bitumen–mineral composition, contact angle, surface tension, adhesion, mathematical modeling, surfactant, polymer, bitumen modification

## Abstract

The purpose of this study was to optimize the processes of wetting fillers by varying the content of such additives as a surfactant and polymer in bitumen–mineral compositions in order to achieve optimal performance. The cosine of the contact angle was used as a criterion for assessing the adhesion of the bitumen binder to the surface of crushed stone. The effect of the additives’ concentration on surface tension and adhesive efficiency in binary and ternary bitumen compositions was studied. The following chemicals were used as additives: the original product AS-1, industrial additive AMDOR-10, and used sealant AG-4I, a product based on polyisobutylene and petroleum oils. AS-1 was obtained from the oil refining waste in the laboratory of M. Kozybayev North Kazakhstan University. The ternary “bitumen–AG-4I–AS-1” composition provided a maximum decrease in the contact angle by 15.96° (gray crushed stone) and by 14.06° (red crushed stone) relative to original bitumen, providing better wettability of the mineral filler particles with the bitumen, and as a result, maximum adhesion between the bitumen and crushed stone. The optimal performance of the bitumen–mineral composition was recorded with the joint presence of additives in the bitumen: AS-1 at a level of 1.0 g/dm^3^ and AG-4I at a level of 1.0 g/dm^3^.

## 1. Introduction

The problem of the quality of asphalt concrete pavements is relevant from the standpoints of the resource-saving and socio-economic competitiveness of any country in the world. These and other objective factors necessitate further improvement in the quality of road bitumen through the search for new scientific and technological solutions to achieve high physical and mechanical characteristics of asphalt concrete pavements and ensure their durability. One of the main reasons for the premature destruction of road surfaces is the low quality of road bitumen [1,2]. Bitumen does not have the necessary ratio of adhesive–cohesive properties, and the asphalt concrete pavements formed on their basis are not capable, under conditions of a constant increase in traffic and freight traffic, of providing the required performance properties [3,4,5,6,7].

According to global practice [8,9,10,11,12,13], an effective way to improve the quality of a bituminous binder is its modification. The most well-known and widely used adhesive action modifiers in road construction are surface-active substances (surfactants), most often cationic ones [14,15,16,17]. The mechanism of their influence on the properties of bitumen–mineral compositions is based on adsorption processes. The positively charged ammonium center or nitrogen atom is concentrated on the polar surface of the mineral filler, and the non-polar hydrocarbon radical is oriented towards the bituminous medium, creating a transition layer that smoothens out the polarity difference between the two phases. As a result of adsorption, the surface tension decreases, and, according to the Young equation, the surface wetting improves, which ultimately leads to an increase in bitumen adhesion to the surface of the mineral filler (Young–Dupre equation) [18,19].

In the use of surfactants in asphalt concrete compositions, cationic surfactants, diamines and polyamines, are most often used [20,21]. As a rule, cationic surfactants are fatty amines such as imidazolines, amidoamines, and diamines. In Europe, cationic surfactants based on stearic, palmitic, and butyric acids are used. In the United States, additives such as amines and ammonium salts, as well as condensation products of aldehydes with organic amine, organic polyamine, and hydrohalides, are widely used [22]. Gilmore and Kugele [23] improved the adhesion of bitumen to a filler using a mixture of imidazoles, polyamines, alkoxylated polyamines, aminocarboxylic esters of amidoamines, while Treybig and Chang [24] developed hydrocarbyl nitrogen adhesive additives containing aromatic heterocyclic compounds, aldehydes, ketones, and amines. Meanwhile, the synthesis of cationic surfactants using petrochemical production wastes, which contain a mixture of higher and lower aldehydes, makes it possible to organize the import-substituting production of our own effective adhesive additives. However, surfactants, as shown by numerous studies [25,26,27,28,29,30], cannot significantly improve some of the physical and mechanical properties of bitumen, especially elasticity and strength.

The scientific direction aimed at improving the physical and mechanical properties of bitumen is the use of polymer additives [25,26,27,28,30]. It has been proven that the efficiency of bitumen modification in each individual case depends on the quantitative ratios of the polymer and bitumen, their compatibility, as well as the temperature regimes for the preparation of the polymer–bitumen binder (PBB) [30]. Thus, by changing the type and concentration of the polymer, it is possible to obtain composite materials with a controlled set of physical and mechanical properties. Currently, the following types of polymers have found application as a polymeric bitumen modifier: elastomers—SBS (styrene–butadiene–styrene) [31], NBR (nitrile butadiene rubber) [32], thermoplastics—EVA (ethyl vinyl acetate) [33,34], EMA (ethylene methyl acrylate) [34], APP (atactic polypropylene) [35], PE (polyethylene) [36], and thermosetting resins [37]. Projects have been implemented (Canada, Kazakhstan) for the modification of bituminous binders with recycled polyethylene waste [38]. The compatibility of bitumen with thermoplastic elastomers (DST-30, IST-30), butyl rubber (BK-2045T), and ethylene–propylene rubber (SKEP M-40 and M-60) has been studied. It has been shown that stable polymer bitumen compositions can be obtained using mixtures with paraffin–naphthenic oils [39]. It has been proven that polymer additives APP (atactic polypropylene) and DST-30-01 (divinylstyrene thermoplastic) improve the wear resistance and anti-deformation properties of bitumen and asphalt concrete [40]. However, despite the attractiveness of this direction of modification, it is economically disadvantageous. The limiting factor for their widespread introduction is the high cost of polymers, as well as the complexity of technological processes. Thus, the combination of a lumpy elastomer with bitumen in the mixer requires a preliminary plasticization of the polymer on rollers; the introduction of polymers in the form of crumbs, granules, and powders requires a high temperature for their melting in the mixer. In this regard, the use of used sealant AG-4I, which is used in large quantities for the anti-corrosion protection of the inner surface of tanks at energy facilities, oil-and-gas-producing enterprises, machine-building plants, urban utilities, and other industries in heating systems, becomes relevant [41]. A solution of polyisobutylene in mineral oil is easily combined with bitumen and gives homogeneous mixtures. This makes it possible to implement a one-stage production scheme, when the polymer and surfactant are added simultaneously, with an intensively operating mixer, into a container with a molten binder (t = 130 °C).

By varying the polymer and surfactant added into the bitumen composition, it is possible to localize the influence of additives within the required modification effects [4]. The improved ability of the modified bitumen to wet the surface of acidic and basic crushed stone will result in a high degree of adhesion between the bitumen and the mineral filler.

Making science-based decisions both on determining the nomenclature and dosing of additives, and on predicting their combined effect in modified bitumen–mineral compositions is extremely important for the development of the modern scientific direction in the field of modifying composites. The use of probabilistic–deterministic modeling (PDM) methods allows for solving this problem and developing recipes with the required characteristics [4,42,43,44,45,46,47,48].

The purpose of this study is to optimize the processes of wetting fillers by varying the content of surfactant and polymer in the bitumen–mineral compositions in order to achieve optimal performance. This makes it possible to obtain modified bitumen–mineral compositions with high rates of adhesion of the binder to the surface of the substrate.

To achieve the goal, the following tasks were set:Investigate the influence of the concentration of additives on the surface tension (*σ_l-g_*) in binary “bitumen–surfactant”, “bitumen–AG-4I”, and ternary “bitumen–surfactant–AG-4I” systems;Study the effect of the concentration of additives in bitumen on the wetting processes of mineral fillers of various types;Determine the thermodynamic work of adhesion and the adhesive effectiveness of modifiers in the composition of a bituminous binder in relation to the surface of mineral fillers;Derive mathematical models of wetting processes based on the established physical and chemical laws of modification.

## 2. Materials and Methods

### 2.1. Materials

To conduct the research, the following materials were used:Oxidized bitumen with penetration 90/130 (bitumen brand BND 90/130—oil bitumen for roads in Kazakhstan), manufactured by Gazpromneft-Bitumen Kazakhstan LLP, Shymkent, Kazakhstan.

The composition and structural characteristics of the samples were determined through IR spectroscopy.

In the IR spectra of the bituminous binder (Figure 1), bands are observed corresponding to the stretching vibrations of -CH_2_ groups in the benzene ring (2930–2910 cm^−^^1^) in the region of 2920 cm^−^^1^ and in the aliphatic chain (2860–2850 cm^−^^1^) in the region of 2851 cm ^−^^1^, as well as deformation vibrations of -CH_2_ and -CH_3_ groups (1385–1370 cm^−^^1^) in the region of 1376 cm^−^^1^. Out-of-plane C-H bending vibrations in aromatic structures are represented by peaks in the low-frequency region of 868 cm^−^^1^ and 810 cm^−^^1^ (810–750 cm^−^^1^, 900–860 cm^−^^1^).

2.Modifying additives:

-AS-1 [49] is a product of the amination of distillation residues of petrochemistry (KON-92), which is a mixture of amines of the general formula
R’-NH_2_, R’-NH-R’’,
where R’ is n-butyl, R’’-2-ethyl-2-hexenyl.

In the IR spectra of the obtained compounds (Figure 2), a peak at 733 cm^−1^ is observed, corresponding to the deformation vibrations of -NH_2_ of primary amines (650–900 cm^−1^), as well as specific absorption bands associated with vibrations of N-H bonds, in the region of 1065 cm^−1^ and 1102 cm^−1^ (1020–1220 cm^−1^). The presence of a peak in the region of 1734 cm^−1^ indicates vibrations of the C=O carbonyl group (1650–1780 cm^−1^). The absorption of methyl fragments is fixed in the region of 2873 cm^−1^ (2885–2860 cm^−1^); stretching vibrations of -CH_2_ groups are presented in the region of 2932 cm^−1^.

-AG-4I is a used sealant, a product based on high-molecular-weight polyisobutylene (PIB) and petroleum oils (manufactured by “Germetika Research and Production Company”, Moscow, Russia).

In the IR spectra of the used sealant (Figure 3), absorption bands are observed corresponding to the deformation vibrations of the C-H bonds in the region of 722 cm^−1^. The doublet at 1366 and 1376 cm^−1^ corresponds to the symmetrical bending vibration of both methyl groups. The cleavage is caused by resonance and is characteristic of the dimethyl-substituted chain. Peaks in the regions of 2853, 2922, and 2953 cm^−1^ indicate stretching vibrations of -CH_2_.

-AMDOR-10—a mixture of polyaminoamides and polyaminoimidazolines (manufacturer CJSC “Amdor”, Saint-Petersburg, Russia), a condensation product of polyamines and higher fatty acids.

The IR spectrum of the adhesive additive AMDOR-10 (Figure 4) is characterized by the following absorption bands: 722 cm^−1^—deformation vibrations of -CH_2_ groups, peak at 733 cm^−1^, characteristic of primary amines (900–650 cm^−1^). The presence of peaks in the region of 1550 cm^−1^ indicates bending vibrations of -NH groups in secondary amides; absorption in the region of 1647 cm^−1^ proves the presence of the C=N bond characteristic of imidazolines (at 1650 cm^−1^). Stretching vibrations of -NH groups are characterized by absorption in the region of 3281 cm^−1^.

3.The following materials were used as substrates for determining the contact angle:-Polymethyl methacrylate (PMMA), as a solid surface standard with a predominance of basic centers (manufacturer “JiangmenKunxin New Material Technology Co., Ltd.”, Omsk, Russia);-Polyvinyl chloride (PVC), the standard of a solid surface with a predominance of acid sites (manufactured by “VitaChem” LLC, Dzerzhinsk, Russia);-Gray crushed stone (plagiogranite);-Red crushed stone (alaskite).

Gray crushed stone is a granite mineral material mined by “KazRosResurs” LLP, Russia (Ornek deposit), the chemical composition of which is presented in Table 1.

Red crushed stone is a granite mineral material mined by “Zhas-Kala” LLP, Kazakhstan (Leningradskoe deposit), the chemical composition of which is presented in Table 2.

### 2.2. Preparation of Modified Bituminous Compositions

The preparation of modified bitumen compositions was carried out in a porcelain container equipped with an automatic stirrer, a thermometer, and electric heating. A total of 100 g of bitumen was heated with constant stirring to 80 °C, achieving a mobile state, after which the temperature was raised to 130 °C (the temperature of mixing this brand of bitumen with mineral material) to get as close as possible to real production conditions. After keeping the bituminous binder in a thermostated mode for 30 min (for heating and removing air bubbles from the bulk phase), the modifier was dosed with continuous stirring, varying its quantitative content from 0.5 to 2.0 g/dm^3^. The bitumen–additive binary composition was stirred at 130 °C for 40 min. This mixing time is sufficient to achieve equilibrium interfacial surface energy and a contact angle of wetting, which was established from the results of preliminary kinetic studies on *σ_l-g_* and cos *θ* in modified bitumen compositions.

Ternary compositions “bitumen–polymer–surfactant” were prepared through the successive introduction of polymer and surfactant modifiers. First, a sealant solution (C = 0.5–2.0 g/dm^3^) was introduced into the molten bitumen (t = 130 °C); then, the composition was stirred in a thermostatically controlled mode for 40 min. After this period of time, AS-1 (C = 0.5–2.0 g/dm^3^) was dosed into the bitumen–AG-4I binary system with a fixed sealant content, keeping the composition under constant stirring and at 130 °C for 40 min to achieve an equilibrium state of the system.

### 2.3. Determination of the Surface Tension of Bituminous Compositions

The measurement of *σ_l-g_* (t = 130 °C) was carried out using the Easy Drop hanging drop method on an automatic setup of the ACAM series (Apex Instruments Co. Pvt. Ltd., Jadavpur Kolkata, India) (Figure 5). The drop was formed using a 0.25 mL dosing syringe Gastight, 1725 RN, Hamilton (Hamilton Central Europe S.R.L., Romania) by dispensing the drop as much as possible from the needle capillary until detachment.

The drop image (Figure 6) was scanned on the monitor screen and processed using the ApexAcamSoftware software (version 2.026.088.1). This software of the installation allows for calculating the surface tension using the Young–Laplace method [50].

A fixed image of a hanging drop allows us to analyze the contour of the drop profile for its shape using the Young–Laplace Equation (1). This equation describes the pressure difference (Laplace pressure) between areas inside and outside a curved liquid surface/interface with basic radii of curvature *r*_1_:(1)$$∆p=(\frac{1}{r_{1}}+\frac{1}{r_{2}})\sigma$$

To calculate the surface tension based on differential geometry, an analytical expression is defined:(2)$$\frac{dF}{ds}=2k-\frac{z∆Pg}{\sigma }-\frac{\sin F}{x}$$
where *F* is the angle between the tangent at the point and the *X* axis; *s* is the length of the arc along the contour; *z* is the coordinate along the vertical axis; *k* is the reciprocal of the radius of curvature; *g* is the free-fall acceleration.

The solution of this equation makes it possible to calculate the surface tension of the studied compositions. An automatic calculation of surface tension was performed for several hundred pairs of points in less than one second. Such a rate makes it possible to trace the change in surface tension (non-equilibrium surface tension) over relatively short periods of time. Surface tension was determined through five individual measurements.

### 2.4. Measuring the Contact Angle

Wettability, the interaction of liquid with a surface, is crucial in many technological processes. The contact angle is the angle at the interface between liquid, air, and a solid, and its value is a measure of the probability of wetting the surface with liquid. The lower the wetting angle, the higher the tendency of the liquid to spread over a solid surface. High values of the wetting angle indicate the tendency of the surface to repel liquid. Measurements of the contact angle of bitumen are used to quantify the free energy of the surface, which provide a fundamental insight into the adhesion and cohesion bonds of a bitumen binder. A static contact angle is determined from a stationary drop [51,52].

#### 2.4.1. Method for Preparing the Surface of Crushed Stone

At the first stage, the surface of two varieties of crushed stone was polished using the disk of a grinding and polishing machine, achieving the same roughness values in order to exclude the influence of this parameter on the contact angle *θ*. The surface roughness *R_a_* was determined using a TR-100 roughness meter (GeoNTD LLC, Moscow, Russia). To measure the hardness of the mineral samples on the Rockswell scale (HRC), a MET-UDA combined hardness tester (MET LLC, Saint-Petersburg, Russia) was used. The measurement results are presented in Table 3.

Microimages of the surface relief of two varieties of polished crushed stone obtained using the semi-contact method of atomic force microscopy with a SOLVER NANO scanning probe microscope (NT MDT LLC, Russia) are shown in Figure 7.

At the second stage of surface preparation, the crushed stone samples were boiled in distilled water for 30 min; then, the crushed stone was dried by hanging vertically in an oven at a temperature of 150 °C, and then cooled at a temperature of 25 °C for 15 min.

#### 2.4.2. Contact Angle Measurement

Measurement of *θ* (t = 130 °C) was carried out using an automatic system for measuring the dynamic contact angle and free surface energy of the ACAM series (Figure 5).

A drop of bitumen was placed on the substrate surface using a dispenser at a bitumen temperature of 130 °C. To obtain an objective image of a bitumen drop, the needle tip was centered horizontally and vertically, and an absolutely white background was set by adjusting the illumination intensity. The shadow image of the drop was obtained using a high-speed camera.

By injecting and pumping out the test liquid, we achieved the optimal drop volume at the set drop rate from the syringe, setting the necessary program parameters. A drop of a given volume was transferred to the substrate surface (Figure 8).

Static contact angles at steady state were measured 20 s after dosing, as recommended in [53]. The contact angle was determined through five individual measurements. The measurement accuracy was ±0.050°.

#### 2.4.3. Method of Probabilistic–Deterministic Planning

The modeling of the effect of modifiers on the process of wetting of mineral fillers was carried out within the framework of the method of probabilistic–deterministic planning (PDP). The research work using PDP consisted of several stages:Determination of factors and levels of their variation.Constructing an experimental plan in the form of a plan matrix consisting of m columns corresponding to the number of input parameters (factors) and n rows corresponding to the number of variations in the given levels (numerical value) of the factors. To ensure the orthogonality of the plan matrix, each level of one input parameter was specified only once with each level of another input parameter.Conducting an active experiment according to the generated plan matrix and establishing the numerical values of the response function (output parameter).Sampling the response function for each level of each factor.Construction of partial dependencies of the response function on each factor.Approximation of particular dependencies and derivation of a generalized mathematical model.

The main factors (input parameters) were determined: the content of the amine derivative AS-1 (C_AS-1_, g/dm^3^: 0–2.0) and the additive AG-4I (C_AS-1_, g/dm^3^: 0–2.0) in the bituminous binder. The numerical values of the levels for each factor are presented in Table 4.

Experiments were carried out on the basis of the constructed orthogonal, multilevel plan-matrix of a two-factor experiment (Table 5).

The cosine of the contact angle (*cos θ*), which was determined experimentally, was taken as the response function.

After the implementation of the active experiment (Table 5), the experimental array was sampled for each level of each factor according to Table 6.

On the basis of a sample of the experimental data array (Table 6), partial dependences of the response function on the content of modifiers were plotted.

At the last stage, partial dependencies were approximated to obtain one-parameter equations characterizing the influence of each factor separately on the response function. To build a multifactorial statistical mathematical model (generalized equation), we used the formula proposed by Protodyakonov, which, in the case of a two-factor experiment, takes the form:(3)$$y=\frac{f\left(x_{1}\right)f(x_{2})}{g_{av}}$$
where *f*(*x*_1_), *f*(*x*_2_)—dependence of the response function on factor *x*_j_; *g_av_*—average value of the actual value of the output parameter for all *n* experiments (general average).

The values of *g_av_* were calculated using the formula:(4)$$g_{av}=\frac{\sum y_{i}}{n}$$
where *∑y_i_* is a set of experimental data in a matrix; *n* is the total number of experiments in the plan matrix.

The estimation of the accuracy of the obtained approximated equations and the resulting mathematical models was evaluated using the coefficients of nonlinear multiple correlation (*R*) and significance (*t_r_*), which were calculated using Equations (5) and (6):(5)$$R=\sqrt{1-\frac{(n-2)\sum {{(y}_{ex}-y_{t})}^{2}}{(n-1)\sum {{(y}_{ex}-y_{av})}^{2}}}$$
(6)$$t_{R}=\frac{R\sqrt{\left(n-2\right)}}{1-R^{2}}$$

#### 2.4.4. Evaluation of the Operational Properties of the Coating (Contact with Water, Elevated Temperature) by the Adhesive Effectiveness of Modifiers

The method of studying the adhesive properties of modified bitumen, currently used, is based on the retention of bitumen-coated mineral material in water with a visual assessment of the surface area from which the bitumen is exfoliated [54]. Visual assessment quite often leads to errors, especially in the case of dark-colored mineral materials; therefore, a quantitative method for determining the adhesion index was used in this work, which made it possible to more objectively assess the adhesive strength of bitumen adhesion to mineral materials [55]. The mineral filler was washed with distilled water and dried at 100 °C in a drying cabinet, the mass was fixed, and crushed stone was introduced into the heated bitumen. The mixture was mixed until the surface of the mineral filler was completely covered with bitumen. After complete coating, the crushed stone was cooled to room temperature, and then the samples were kept in boiling water for 30 min. The adhesion index and the adhesive efficiency of modifiers were calculated using the mass of the exfoliated bitumen from the crushed stone surface after boiling.

The bitumen adhesion index *X* (% by weight) was calculated according to formula (7):(7)$$X=\frac{m_{1}-m}{m_{2}}\;.\;100\%$$
where *m*_1_ is the mass of the bitumen–mineral mixture after boiling, g; *m* is the weight of the mineral material, g; *m*_2_ is the weight of bitumen before boiling, g.

The adhesive efficiency of additive *A* (%) was calculated using Formula (8).
(8)$$A=\frac{X_{i}-X_{o}}{X_{o}}\;.\;100\%$$
where *Xi* is the adhesion index of bitumen modified by the additive (% by weight); *X_o_* is the adhesion index of the initial bitumen (% by weight).

## 3. Results and Discussion

### 3.1. Surface-Active Properties of Binary Systems “Bitumen–Additive” at Interphase Boundaries with Air

The presented isotherms of surface tension (t = 130 °C) clearly reflect the dynamics of the change in σ at the “bitumen–air” boundary with the introduction of additives into bitumen (Figure 9).

Judging by the surface tension isotherm (Figure 9), the surface-active properties of bituminous compositions with a solution of polyisobutylene in mineral oil are determined by complex associative–dissociative transformations in the micellar structure of bitumen (the core is asphaltenes, the shell is resins, the medium is oil). Thus, the introduction of a limited concentration (C ≤ 1.0 g/dm^3^) of a sealant into bitumen leads to a weakening of intermolecular interactions in bitumen macroassociates and the release of their surfactant components. The presence of unbound amphiphilic molecules in the “bitumen–AG-4I” binary system stimulated their concentration in the surface layer, which was confirmed by a decrease in the values of *σ_l-g_* from 45.39 to 41.31 mN/m. However, with a further increase in the concentration of the additive (C > 1.0 g/dm^3^), the progress of the reverse process, association, was noted. This process of polymer association with bitumen macroassociates was accompanied by a spontaneous loss of free surfactant molecules and a corresponding continuous deterioration in surface properties (ascending section of the isotherm). At C = 2.0 g/dm^3^, the surface tension turned out to be at a level close to unmodified bitumen (*σ* = 45.56 mN/m). This indicates that there are no free surfactant molecules at the interface with air and, at the same time, their localization in the internal structure formed in the volume of the dispersed system of intermolecular associates.

The extreme nature of the patterns of change in *σ_l-g_* was also recorded when AMDOR-10 was introduced into bitumen compositions (Figure 9), which is widely used for asphalt concrete pavements in Kazakhstan. The use of this polymer additive, but in contrast to the solution of polyisobutylene (Figure 9), an amphiphilic compound (a mixture of polyaminoamides and polyaminoimidazolines), introduces a number of its own features. The high-molecular variety of nitrogen-containing additives is characterized by a smaller decrease in surface tension, as well as a shift of the extremum to the region of lower concentrations. The maximum depression of surface tension (Δ*σ* = 1.02 mN/m) was noted in bituminous compositions with AMDOR-10 at C = 0.5 g/dm^3^. The dosing of the additive above 0.5 g/dm^3^ contributed to its concentration in volume, but not at the interface with air, and, as a result, to an increase in surface tension. With an increase in the concentration of AMDOR-10 from 0.5 to 2.0 g/dm^3^, the surface tension increased by 0.85 mN/m, i.e., it practically returned to its original state (*σ* = 45.22 mN/m). The change in surface tension for bitumen compositions with low-molecular varieties of the nitrogen-containing additive AS-1 (Figure 9) demonstrates the traditional form of surfactant isotherms that are not prone to association processes: as the concentration increases from 0 to 1.0 g/dm^3^, the surface tension decreases from 45.39 to 40.8 mN/m and remains unchanged outside this concentration range (C > 1.0 g/dm^3^).

For a comparative assessment of the ability of additives to reduce the specific surface energy of bitumen at the boundary with air, the surface activity index was used:(9)$$g={\left(\frac{d\sigma }{dc}\right)}_{c\to 0}$$

The surface activity of surfactants (*dσ/dc*) was calculated from the tangent of the slope of the linear sections of the surface tension isotherms (Figure 9). The numerical coefficients of the linear sections of the surface tension isotherms were determined using the least squares method. The values of the correlation coefficients were not lower than 0.99.

The indicators of the surface activity of modifiers AG-4I, AMDOR-10, and AS-1 at the interface between bitumen and air are presented in Table 7.

A comparative analysis of the values of the surface activity of the studied binary systems “bitumen–additive” shows (Table 7) that the low-molecular-weight AS-1, which is characterized by a smaller size of the hydrocarbon radical (M = 250 a.m.u.), in comparison with the high-molecular-weight AMDOR-10 (M = 2260 a.m.u.), exhibits maximum surface activity at the interface with air. In bituminous compositions, the surface activity of AS-1 is 5.16 mN×dm^3^/m×g; for AMDOR-10. this value is 2.5 times lower—2.04 mN×dm^3^/m×g (reversal of the Duclos–Traube rule in non-polar media). Bituminous surfactants activated by the introduction of AG-4I (*g* = 4.08 mN×dm^3^/m×g) occupy an intermediate position between two types of nitrogen-containing additives in terms of surface activity at the interface with air.

Based on the achievement of maximum surface activity values, AG-4I and AS-1 were used in the mixed composition, including the joint presence of two modifiers in bitumen.

### 3.2. Surface-Active Properties of Ternary Systems “Bitumen–Polymer–Surfactant” at the Interface with Air

The results of the experimental studies of surface tension (*σ_ex_*) in the mixed composition, including the joint presence of AG-4I and AS-1, are presented in Table 8. Additionally, for a comparative assessment of the surface properties of additives in binary and ternary systems, a comparison of the *σ_ex_* and *σ_c_* values was used. Surface tension (*σ_c_*) was calculated as an additive value:(10)$$\sigma_{c}=\sigma_{0}-({∆\sigma }_{AG-4I}+∆\sigma_{AS-1}),$$
where Δ*σ_AG-_*_4*I*_ and Δ*σ_AS-_*_1_ are the change in surface tension in the binary system relative to unmodified bitumen.

The data obtained (Table 8) indicate the closeness (within the statistical error) of the experimental and calculated values of surface tension (*σ_ex_* = *σ_c_* ± 0.05 mN/m). This indicates the absence of intermolecular interactions and spatial complications between the components of the compositions.

Thus, it can be stated that the change in the specific surface energy is an additive value that takes into account the individual contributions of both AG-4I and AS-1. It should also be noted that the concentration threshold of additives to achieve the minimum *σ_l-g_* remained the same (C = 1.0 g/dm^3^) as in individual compositions. With the joint introduction of AG-4I (C = 1.0 g/dm^3^) and AS-1 (C = 1.0 g/dm^3^), the surface tension decreased to 37.20 mN/m compared to unmodified bitumen (Δ*σ* = 8.19 mN/m).

### 3.3. Investigation of the Processes of Wetting of Solid Surfaces by Binary Systems “Bitumen–Additive”

The results of the measurements of the contact angle *θ* (Table 9) indicate that the introduction of additives into bitumen stimulates the process of wetting a solid surface. However, the nature of the development of this process is determined not only by their quantitative content, but it is also clearly delineated depending on the characteristics of the nature of the solid surface and the qualitative composition of surfactants.

The maximum wetting effect of AG-4I, AMDOR-10, and AS-1 on the surface of gray crushed stone (Table 9) coincide with the concentration of 1.0 g/dm^3^. A comparison of the contact angles at a given concentration (C = 1.0 g/dm^3^) showed that the greatest decrease in *θ* occurred in the presence of AG-4I. Thus, with an increase in the content of AG-4I in bitumen from 0 to 1.0 g/dm^3^, the contact angle decreased by 13.16° (relative to the base case without AG-4I) and amounted to 112.50°. When bitumen was modified with nitrogen-containing surfactants (C = 1.0 g/dm^3^), a smaller decrease in the contact angle was recorded—decrease of 8.86° (AMDOR-10) and 10.56° (AS-1).

In relation to the red crushed stone for binary compositions “bitumen–AG-4I” and “bitumen–AMDOR-10”, we noted a different amplitude of changes in the contact angle (Table 9). The *θ* values decreased by 8.57° (AG-4I) and 4.27 (AMDOR-10), which is 1.5 and 2.0 times less than on the surface of the gray crushed stone, respectively. And only in the presence of AS-1 did the decrease in *θ* have values close to those of the gray crushed stone’s surface (Δ*θ* = 10.57° at C = 1.0 g/dm^3^). Outside these concentration areas (C > 1.0 g/dm^3^), the contact angle changed insignificantly both on the surface of the gray crushed stone and on the surface of red one.

For a comparative assessment of the wetting activity of modifiers, the linear sections of wetting isotherms (Figure 10) were approximated using Equation (11) of the form:(11)$$cos\;\theta =bC+cos{\;\theta }_{0}$$
where *cos θ_0_*—cosine of the contact angle formed on the surface of the crushed stone by a bituminous binder without a modifying additive; *b* is a constant.

Differentiating Equation (11), we obtained the following:(12)$$a={\left(\frac{dcos\theta }{dc}\right)}_{c\to 0}$$

The corresponding values of a are presented in Table 10.

A comparative analysis of the values of the wetting activity of three types of additives showed (Table 10) that in the process of the oleophilization of an energetically inhomogeneous surface of the gray crushed stone, bituminous compositions with AG-4I (*a* = 0.19 dm^3^/g) showed the maximum wetting activity. For compositions with AMDOR-10, this indicator was 1.6 times lower and amounted to 0.12 dm^3^/g. The wetting activity of AS-1 (*a* = 0.15 dm^3^/g) took an intermediate position among the studied additives. Thus, it can be stated that additives, in terms of wetting activity, in relation to the gray crushed stone, form a series: AG-4I > AS-1 > AMDOR-10.

However, the sequence of this row changes when moving to the surface of red crushed stone. With an increase (Table 1 and Table 2) in the total content of Al_2_O_3_, CaO, MgO, and Na_2_O by 3.19% and a decrease in the chemical composition of the crushed stones SiO_2_ (by 2.17%), Fe_2_O_3_ (by 0.38%), and K_2_O (by 0.64%), the wetting activity for the same binary compositions with AG-4I was reduced by 0.06 dm^3^/g and was 0.13 dm^3^/g (Table 10). As a result, additives form a different series of wetting activity on the surface of red crushed stone: AS-1 > AG-4I > AMDOR-10.

The observed differences in the wetting activity of the additives can be explained by the factor of localization of polar groups on the active sites of the solid surface in accordance with the principle of acid–base interactions. As shown by chemical analysis (Table 1 and Table 2), the surface of the studied mineral fillers consists of separate sections (domains) of different chemical compositions, which include oxides, both acidic and basic. According to the acid–base theory, cationic surfactants are concentrated on the acidic sites of a solid surface, and anionic surfactants are concentrated on the basic ones.

To confirm the role of acid–base interactions in the wetting processes, we measured the contact angle *θ* of the same modified bituminous compositions on reference surfaces of an acidic (PVC) and basic (PMMA) nature (Figure 11 and Figure 12).

Following from the results obtained (Figure 11), on the acidic surface of PVC (*R_a_* = 1.9 µm), which has sufficiently strong electrophilic centers [56], nitrogen-containing surfactants with a lone pair of electrons are able to interact with them according to the donor–acceptor mechanism. As a result of adsorption, the values of σl decrease and, according to Young’s equation, surface wetting improves—the contact angles *θ* decrease by 8.53° (AS-1) and by 3.86° (AMDOR-10) at C = 1.0 g/dm^3^. In the case of using AG-4I, the change in *θ* was negligible (Δ*θ* = 1.67° at C = 1.0 g/dm^3^), which indicates that there was practically no adsorption process on the PVC surface. This serves as evidence of the predominantly acidic nature of bituminous surfactants released under the influence of AG-4I, which was previously revealed by the results of measuring *σ_l-g_* in this binary system (Figure 11). On the basic surface of PMMA, which is opposite in nature (Figure 12), where bituminous surfactants can form acid–base bonds, the binary compositions “AG-4I–bitumen” demonstrated the intensification of wetting processes up to C ≤ 1.0 g/dm^3^; the contact angle *θ* decreased by 9.43° in comparison with unmodified bitumen and amounted to 92.08°.

Nitrogen-containing additives in bitumen (Figure 11), upon the appearance of the reference surface of PMMA (*R_a_* = 7.1 µm), which is similar in nature, rush into the volume of the system and are included in the micellar structure of bitumen, providing the entire system with a decrease in surface activity, as a result of which *σ_s-l_* increases. As a consequence, we saw a sharp deterioration in the wetting of the PMMA solid surface. As the concentration of AS-1 increased (from 0.5 to 2.0 g/dm^3^), the contact angle *θ* increased in comparison with unmodified bitumen (*θ* = 101.51°) by 34.28° (Figure 12). The effect of reducing the polarity of bitumen due to the introduction of AMDOR-10 was much lower; *θ* increased by only 9.27–9.71° (Figure 12).

Thus, the studies carried out on the reference surfaces show that nitrogen-containing surfactants (AS-1, AMDOR-10) were localized on the acidic centers of the mosaic surface of the crushed stone, and acidic surfactants delocalized from the composition of bitumen by the sealant AG-4I are located on the basic centers. At the same time, the interaction on the crushed stone surface was determined by the concentration of the surface adsorption centers of a certain type and their strength, as well as the type and concentration of the adsorbed substance.

As a result of the local adsorption of the surfactant components of bituminous compositions, there was a general decrease in the specific surface energy at the “solid–liquid” interphase boundary (*σ_s-l_*), which estimated indirectly. In accordance with the well-known Young equation, in the case of the contact of a liquid drop with the surface of a solid body, at equilibrium, the following equality takes place:(13)$$\sigma_{l-g}cos\;\theta =\sigma_{s-g}-\sigma_{s-l}=∆\sigma$$

Based on the fact that the specific surface energy *σ_s-g_* at the boundary with air does not depend on the surfactant concentration (*σ_s-g_* = const), the product of the experimentally determined values *σ_l-g_* and *cos θ* showed by how much the specific surface energy *σ_s-l_* changed with the input concentration of additives (Table 11) with respect to some constant value *σ_s-g_*.

Additionally, for a comparative assessment of these processes, the relative changes in the specific surface energy of a solid surface (α) were calculated:(14)$$\alpha =\frac{∆\sigma_{s0}-∆\sigma_{s}}{∆\sigma_{s0}}\times 100\%$$
where Δ*σ_s_*_0_ is the change in the specific surface energy of the mineral filler during the application of unmodified bitumen; Δ*σ_s_*—change in specific surface energy by modified bitumen compositions.

The analysis of the obtained results (Table 11) showed that as the quantitative content of additives in bitumen on the surface of gray crushed stone increased, the decrease in the specific surface energy *σ_s_* increased and reached the minimum values (maximum Δ*σ_s_*) at C = 1.0 g/dm^3^. Judging by the increase in Δ*σ_s_* relative to unmodified bitumen (by 40.37% at C = 1.0 g/dm^3^), adsorption processes were most pronounced in the presence of AG-4I. A compensation for the excess surface energy of the gray crushed stone with nitrogen-containing surfactants was lower in comparison with AG-4I and was 34.90% (AS-1) and 23.59% (AMDOR-10).

The change of the surface mosaic during the transition from gray to red crushed stones means a change in the activity of adsorption centers and their nature. As a result, on the surface of the red crushed stone, the adsorption processes in the presence of both AG-4I and AMDOR-10 significantly decreased. This was evidenced by a smaller decrease in the specific surface energy *σ_s-l_*. So, when AG-4I was introduced into the bitumen, the maximum change in the specific surface energy at the same concentration (C = 1.0 g/dm^3^) was only −19.42 mN/m, and α was almost 1.5 times lower than on the surface of the gray crushed stone. When dosing into AMDOR-10 bitumen, in the entire area of the studied concentrations (C = 0.5–2.0 g/dm^3^), the change in the specific surface energy of the red crushed stone was very weakly expressed (Δ*σ* = −25.50–−24.14 mN/m) and was characterized by minimal values of α (Table 11). And only bituminous compositions with AS-1 retained adsorption patterns close to those of the gray crushed stone, which entailed a similar decrease in the specific surface energy and α values (Table 11). This explains the fact that in the process of the oleophilization of the surface of the red crushed stone, the wetting activity of AS-1 prevailed.

### 3.4. Investigation of the Processes of Wetting of Solid Surfaces by Ternary Systems “Bitumen–Surfactant–Polymer”

The results of experimental measurements of the contact angle *θ* (*θ*_ex_) in mixed compositions, including the joint presence of AG-4I and AS-1, are presented in Table 12.

It follows from the presented data that the introduction of AS-1 into bitumen compositions with a solution of polyisobutylene opened up additional opportunities for intensifying the wetting processes of both the gray and red crushed stones. The addition of AS-1 shifted the contact angle to lower values and reached a minimum at C_AS-1_ = 1.0 g/dm^3^ and C_AG-4I_ = 1.0 g/dm^3^. In these concentration modes, the composition “bitumen–AG-4I–AS-1” caused a decrease in the contact angle by 15.96° (gray crushed stone) and by 14.06° (red crushed stone) relative to unmodified bitumen (Table 12).

The combination of two additives localized on the crushed stone adsorption centers of a different nature led to a deeper decrease in the specific surface energy Δ*σ_s_* (Table 13).

The relative changes in the specific surface energy α were 52.37% (gray crushed stone) and 47.06% (red crushed stone), which were 17.47% (gray crushed stone) and 12.98% (red crushed stone) more than in the binary (Table 11) compositions with AS-1 (C_AS-1_ = 1.0 g/dm^3^). In comparison with the binary compositions “bitumen–AG-4I”, the changes in α amounted to 12.00% (gray crushed stone) and 18.38% (red crushed stone) with C_AG-4I_ =1.0 g/dm^3^.

As a result, with the optimal content of additives (when C_AS-1_ = 1.0 g/dm^3^ and C_AG-4I_ = 1.0 g/dm^3^), the wetting activity of the system “bitumen–AG-4I–AS-1”, calculated by the tangent of the slope of the linear sections of wetting isotherms (Figure 13a,b), increased by 2–3 times compared with the binary systems (Table 10) and was 0.44 dm^3^/g on the surface of the gray crushed stone, and on the surface of the red crushed stone, it was 0.34 dm^3^/g (Figure 13b).

Summarizing the above, we can state that the wetting effect upon the joint introduction of AG-4I and AS-1 into bitumen is determined by the contribution of each additive, which is limited by the number and nature of the adsorption centers of the energetically inhomogeneous surface of mineral fillers.

### 3.5. Evaluation of the Operational Properties of the Coating on the Basis of the Adhesive Effectiveness of Modifying Additives

The established patterns of change in the contact angle and surface tension depending on the quantitative content of the three types of modifying additives should have a decisive influence on the adhesive interaction of bitumen with the surface of mineral fillers. The role of additives in improving one of the fundamental characteristics of the quality of a bituminous binder was confirmed using a direct method for determining adhesion (Table 14). This method is based on the quantitative determination of the exfoliated mass of the bituminous coating on crushed stone after boiling in water (t = 30 min).

The research results presented in Table 14 indicate that the modified bituminous compositions in terms of adhesive efficiency in relation to the gray crushed stone form a series: AG-4I > AS-1 > AMDOR-10.

In comparison with unmodified bitumen, in the presence of AG-4I, the maximum adhesive efficiency (38.67%) was recorded at the same concentration point (C = 1.0 g/dm^3^) as the smallest contact angle *θ* (112.5°, Table 9). The adhesive efficiency of the isoconcentration (C = 1.0 g/dm^3^) bituminous composition in the presence of AS-1 was lower by 9.35% (29.32%) and by 15.81% (22.86%) in the presence of AMDOR-10 compared to AG-4I, which correlates with a smaller amplitude of change in *θ* (Table 9).

The modified bituminous compositions in terms of adhesive efficiency in relation to the red crushed stone formed a series of a different sequence: AS-1 > AG-4I > AMDOR-10. A change in the quantitative ratio of oxides in the chemical composition of the crushed stone led to a different ratio of changes in the contact angle among additives (Table 9) and, as a result, adhesive efficiency. The bituminous composition with AS-1 (C = 1.0 g/dm^3^), while maintaining the proximity of the decrease in *θ* (Table 9), had almost the same value of adhesive efficiency (*A* = 28.76%) with the surface of the gray crushed stone.

A smaller decrease in *θ* (relative to unmodified bitumen) by 1.5–2.0 times (C = 1.0 g/dm^3^) for AG-4I (Δ*θ* = 8.57°) and for AMDOR-10 (Δ*θ* = 4.27°) resulted in lower indicators of adhesive efficiency by 4.51% (A = 24.25%) and 7.70% (A = 21.06%), respectively, in comparison with the isoconcentration bitumen composition with AS-1.

The presented adhesion indicators in the ternary composition (Table 15) indicate that the introduction of AS-1 in the composition of bitumen with a solution of polyisobutylene increased the adhesion of bitumen to the surface of both the gray and red crushed stones.

The addition of AS-1 increased the adhesive efficiency, reaching a maximum at C_AS-1_ = 1.0 g/dm^3^ and C_AG-4I_ = 1.0 g/dm^3^. In these concentration modes, the composition caused an increase in adhesive efficiency by 52.28% (gray crushed stone) and 36.70% (red crushed stone) relative to unmodified bitumen. This is consistent with the previously established deeper oleophilization of crushed stone with a combination of two modifiers localized on different-in-nature adsorption centers of the solid surface: Δ*θ* = 15.96° (gray crushed stone) and Δ*θ* = 14.06° (red crushed stone).

### 3.6. Mathematical Modeling of Wetting Processes

The results of an active experiment aimed at deriving a mathematical model of the process of wetting the surface of two mineral samples of crushed stone with a bituminous binder are presented in Table 16.

A selection of the values of the cosine of the contact angle according to the levels of the concentration factors of the two additives is presented in Table 17.

On the basis of a sample of the experimental data array, partial dependences of the response functions on the input parameters were built (Figure 14).

The generalized models of the process of wetting the surface of two samples of crushed stone have the following form:

Gray crushed stone:(15)$$cos\theta =\frac{\left(-0.0452C_{AS-1}^{2}+0.1096C_{AS-1}-0.4419)(0.0658C_{AG-4I}^{2}+0.1719C_{AG-4I}-0.4732\right)}{-0.4}$$

Red crushed stone:(16)$$cos\theta =\frac{\left(-0.0471C_{AS-1}^{2}+0.1226C_{AS-1}-0.5186)(-0.0487C_{AG-4I}^{2}+0.1224C_{AG-4I}-0.516\right)}{-0.47}$$

The calculations showed satisfactory convergence of the experimental and calculated values of the response function (for the 95th significance level): *R* > 0.72 and *t_R_* > 2.

On the basis of the generalized Equations (15) and (16), nomograms (Figure 15) were obtained, which made it possible to determine the values of the above parameters in order to achieve fixed values of *cos θ*. Thus, the values of *cos θ* = −0.36 were achieved at C_AS-1_ = 1.8 g/dm^3^ and C_AG-4I_ =1.1 g/dm^3^.

Thus, two-factor nomograms were constructed that allow for solving applied problems, in particular, the optimization of bituminous mineral compositions.

## 4. Conclusions

In bitumen systems with a limited concentration of AG-4I (C ≤ 1.0 g/dm^3^), the effect of reducing the surface energy at the interphase boundary with air is achieved due to the concentration of surfactants in the surface layer, which are part of the structure of the bitumen itself, which is facilitated by the development of destructive processes, accompanied by the destruction of associates and the release of active components.Differences in the surface activity of nitrogen-containing surfactants are due to their structural and geometric parameters and molecular weight composition. Their surface activity decreases with increasing molecular weight.In the “bitumen–AG-4I–AS-1” ternary systems, the change in the specific surface energy at the liquid–gas interface was an additive quantity that took into account the separate contribution of both AG-4I and AS-1. The concentration threshold of additives to achieve the minimum *σ_l-g_* remained the same (C = 1.0 g/dm^3^) as in individual compositions. With the joint introduction of AG-4I (C = 1.0 g/dm^3^) and AS-1 (C = 1.0 g/dm^3^), the surface tension decreased to a value of 37.20 mN/m compared to unmodified bitumen (Δ*σ* = 8.19 mN/m).The determining factor in the implementation of the wetting action of additives with respect to mineral fillers is the localization of their polar groups on the active centers of the mosaic solid surface in accordance with the principle of acid–base interactions. Nitrogen-containing surfactants (AS-1 and AMDOR-10) are localized on acid centers, and acidic surfactants released from the micellar structure of bitumen with the introduction of a limited (C ≤ 1.0 g/dm^3^) concentration of AG-4I are localized on basic centers.Additives with wetting activity in relation to the gray crushed stone formed the series AG-4I > AS-1 > AMDOR-10. The maximum decrease in the specific surface energy *σ_s-l_* and the contact angle coincided with the concentration of 1.0 g/dm^3^.Additives in relation to the red crushed stone in a row of decreasing wetting activity formed a different sequence: AS-1 > AG-4I > AMDOR-10. When applying isoconcentration bitumen compositions (C = 1.0 g/dm^3^), the change in the specific surface energy of the red crushed stone was significantly less in the presence of AG-4I (Δ*σ_s_* = −19.42 mN/m) and AMDOR-10 (Δ*σ_s_* = −24.14 mN/m). Bituminous compositions with AS-1 retained the proximity of reducing the specific surface energy (Δ*σ_s_* = −17.00 mN/m). In comparison with unmodified bitumen (*θ* = 126.87°), the decrease in the *θ* values in the presence of AS-1 had indicators close to those of the gray crushed stone surface (Δ*θ* = 10.57°) and was 1.5–2.0 times less for AG-4I (*θ* = 8.57°) and AMDOR-10 (*θ* = 4.27°), respectively, at C = 1.0 g/dm^3^.In the ternary systems, the combination of two modifying additives localized on crushed stone adsorption centers of a different nature led to a deeper decrease in the specific surface energy Δ*σ_s_*. The relative changes in the specific surface energy α were 52.37% (gray crushed stone) and 47.06% (red crushed stone), which was 17.47% (gray crushed stone) and 12.98% (red crushed stone) more than in the binary compositions with AS-1 (C = 1.0 g/dm^3^). The addition of AS-1 shifted the contact angle to lower values and reached a minimum at C_AS-1_ = 1.0 g/dm^3^ and C_AG-4I_ = 1.0 g/dm^3^.The introduction of AS-1 into the composition of bitumen with a solution of polyisobutylene increased the adhesion of the bitumen to the surface of both the gray and red crushed stones. The addition of AS-1 increased the adhesive efficiency, reaching a maximum at C_AS-1_ = 1.0 g/dm^3^ and C_AG-4I_ = 1.0 g/dm^3^. In these concentration modes, the composition caused an increase of 52.28% (gray crushed stone) and 36.70% (red crushed stone) in the adhesive efficiency relative to unmodified bitumen.Based on the results of the mathematical modeling, two-factor nomograms were constructed that allow for solving applied problems, in particular, the optimization of bitumen mineral compositions with predetermined characteristics.

## Figures and Tables

**Figure 1 polymers-16-00714-f001:**
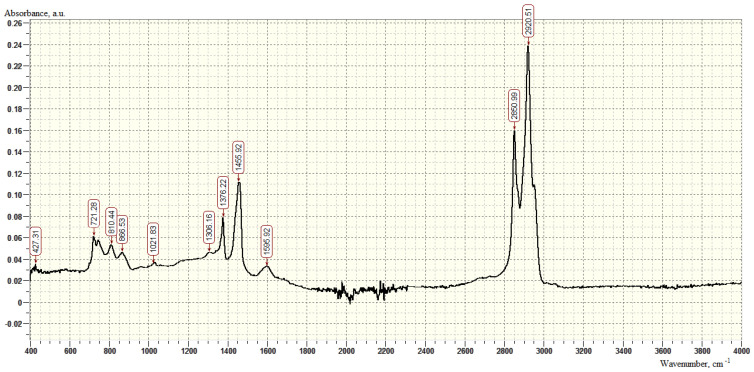
IR spectrum of samples of bituminous binder BND 90/130.

**Figure 2 polymers-16-00714-f002:**
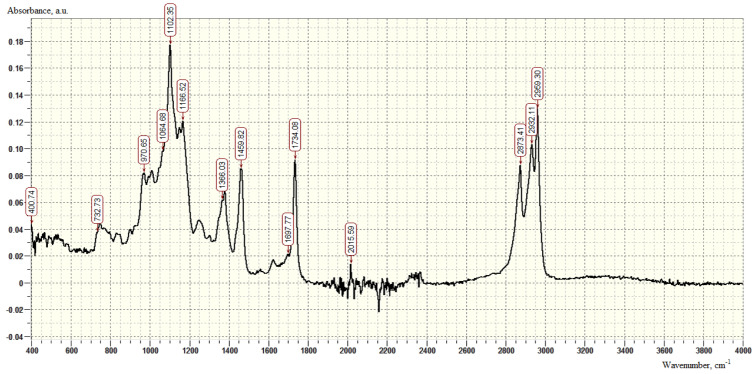
IR spectrum of AS-1 modifier samples.

**Figure 3 polymers-16-00714-f003:**
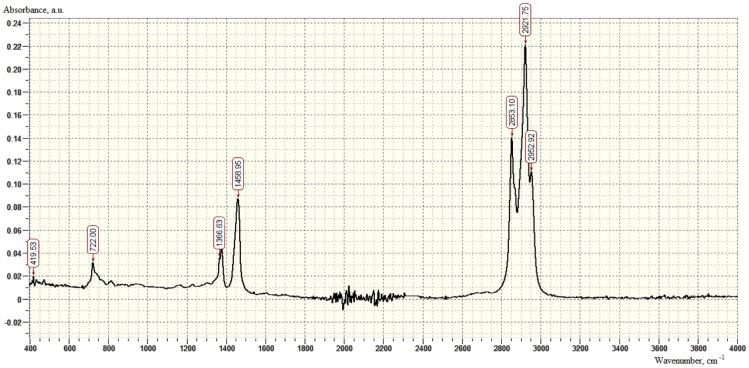
IR spectrum of AG-4I samples.

**Figure 4 polymers-16-00714-f004:**
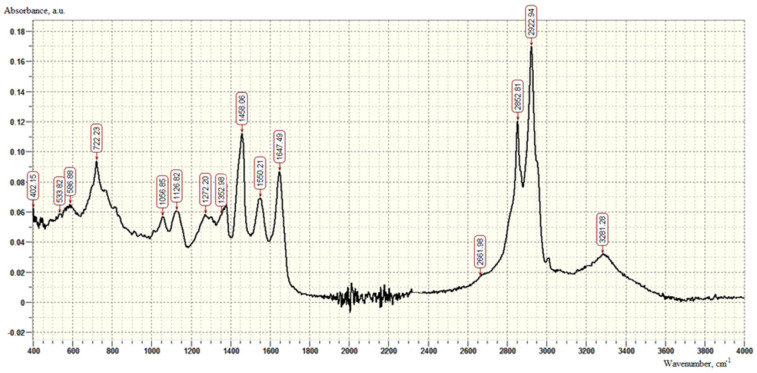
IR spectrum of AMDOR-10 modifier samples.

**Figure 5 polymers-16-00714-f005:**
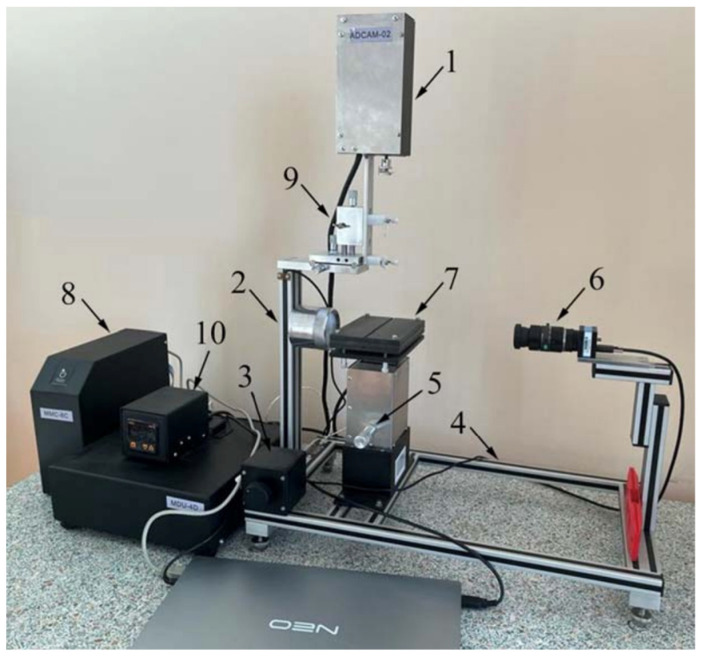
ACAM automatic contact angle measurement system: 1—dispenser module; 2—light source; 3—lighting intensity regulator; 4—base; 5—table raising regulator; 6—camera; 7—table with a substrate; 8—chief controller; 9—horizontal and vertical needle adjustment systems; 10—engine of the dispenser module.

**Figure 6 polymers-16-00714-f006:**
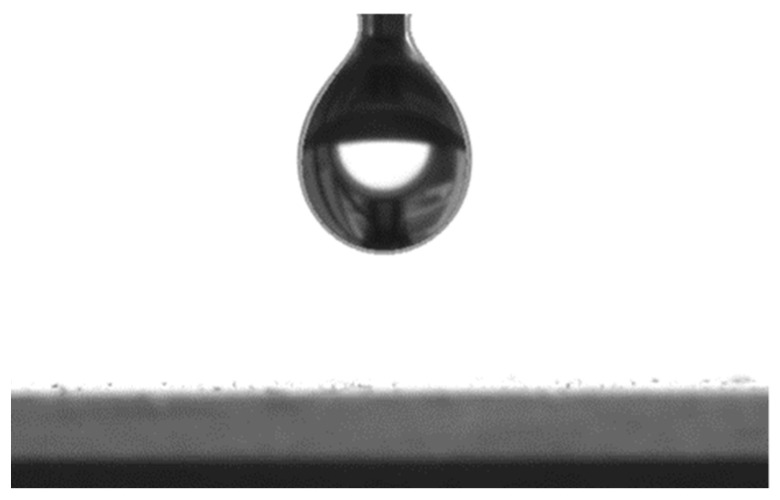
Image of a bitumen drop dosed to determine surface tension.

**Figure 7 polymers-16-00714-f007:**
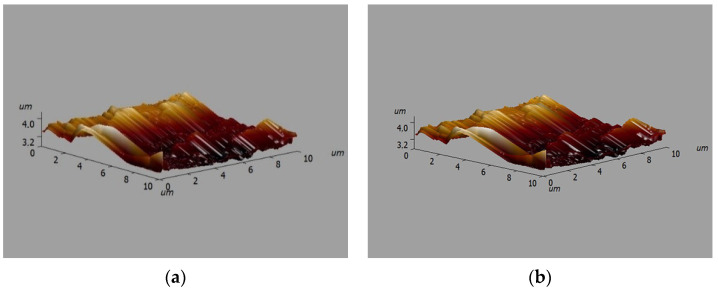
Microimages of the relief of the surface of gray (**a**) and red crushed stones (**b**).

**Figure 8 polymers-16-00714-f008:**
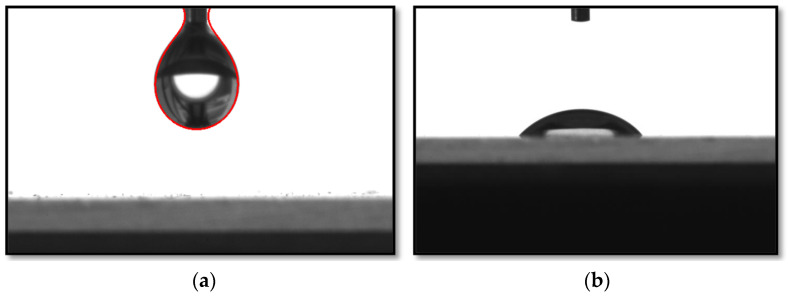
The process of transferring a drop of solution from the tip of the needle to the surface of the substrate (**a**); a drop spread over the surface of the substrate (**b**).

**Figure 9 polymers-16-00714-f009:**
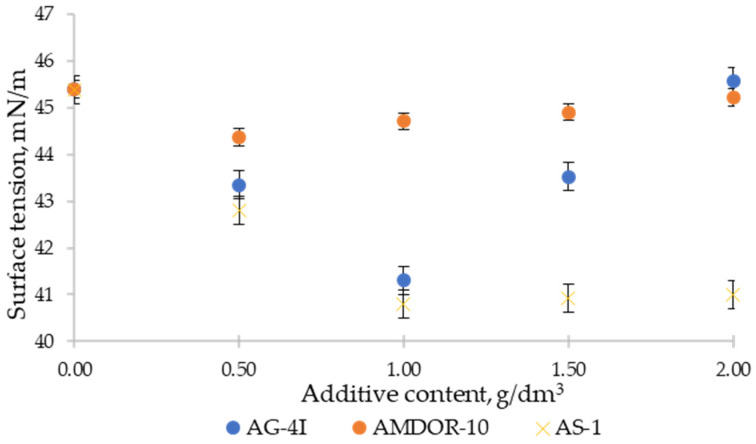
Isotherms (t = 130 °C) of surface tension in the system: “bitumen–additive”.

**Figure 10 polymers-16-00714-f010:**
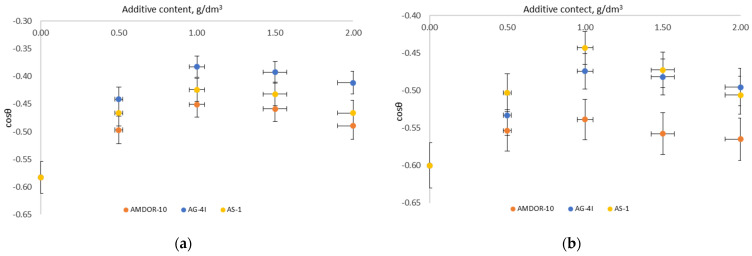
Dependence of the cosine of the contact angle of the surfaces of gray (**a**) and red crushed stones (**b**) on the content of modifying additives in binary systems “bitumen–additive”.

**Figure 11 polymers-16-00714-f011:**
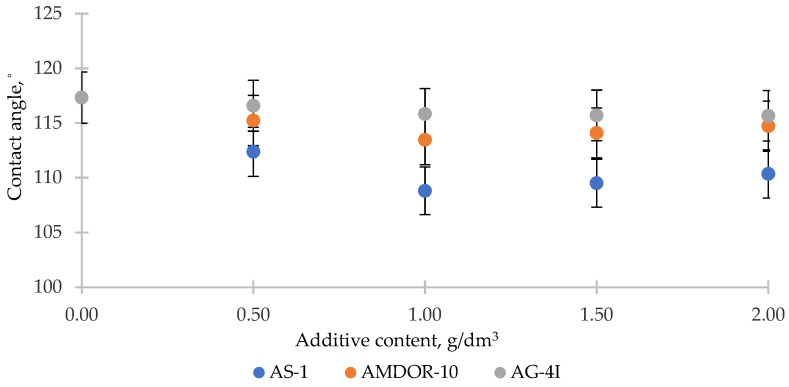
Dependence of the contact angle on the content of the additive on the reference surface of the acidic nature of PVC.

**Figure 12 polymers-16-00714-f012:**
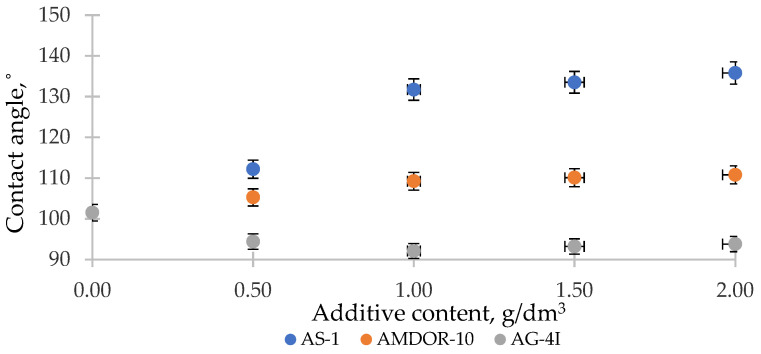
Dependence of the contact angle on the content of the modifying additive on the reference surface of the basic nature of PMMA.

**Figure 13 polymers-16-00714-f013:**
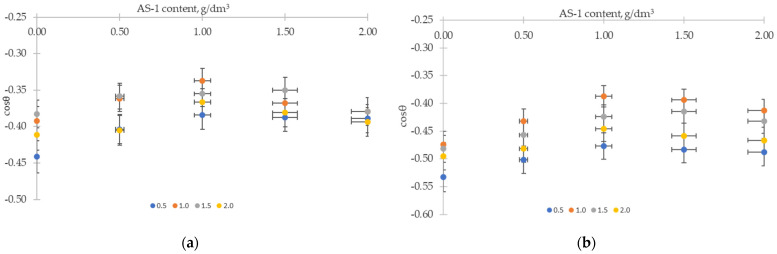
Dependence of the cosine of the contact angle of the surface of gray (**a**) and red crushed stones (**b**) on the content of AS-1 in compositions with a fixed content of AG-4I: 0.5 g/dm^3^; 1.0 g/dm^3^; 1.5 g/dm^3^; 2.0 g/dm^3^.

**Figure 14 polymers-16-00714-f014:**
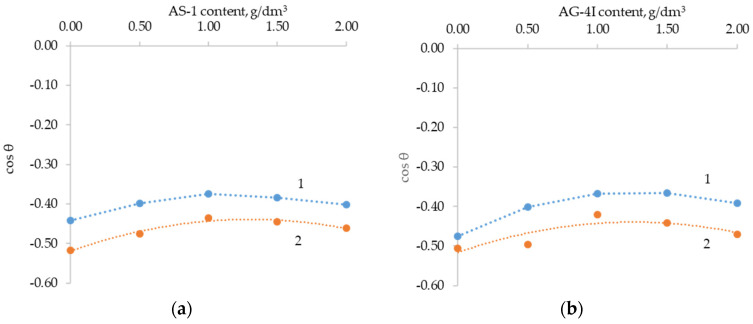
Partial dependences of the change in the cosine of the contact angle on the content of AS-1 (**a**) and AG-4I (**b**): 1—gray crushed stone; 2—red crushed stone.

**Figure 15 polymers-16-00714-f015:**
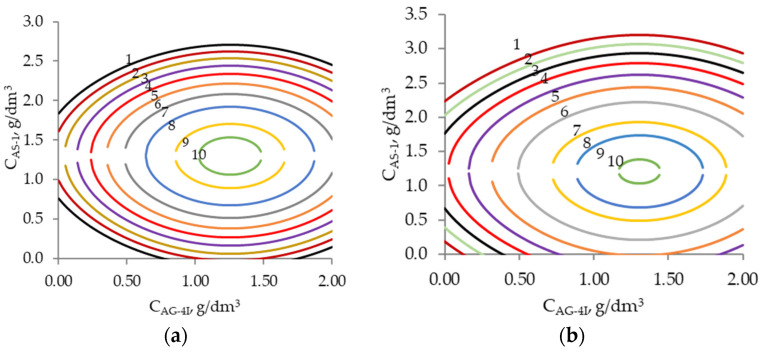
Lines of fixed values of *cos θ* on the content of AS-1 and AG-4I. (**a**). *cosθ*: 1—−0.500; 2—−0.480; 3—−0.460; 4—−0.440; 5—−0.420; 6—−0.400; 7—−0.380; 8—−0.360; 9—−0.350; 10—−0.340. (**b**). *cosθ*: 1—−0.500; 2—−0.490; 3—−0.480; 4—−0.470; 5—−0.460; 6—−0.450; 7—−0.440; 8—−0.43; 9—−0.420; 10—−0.415.

**Table 1 polymers-16-00714-t001:** Chemical composition of the “Ornek” gray crushed stone.

SiO_2_ %	TiO_2_ %	Al_2_O_3_ %	Fe_2_O_3_ %	CaO %	MgO %	S %	Mn ppm	K_2_O %	Na_2_O %
75.37	0.98	12.04	2.67	0.78	0.31	-	774	5.27	2.58

**Table 2 polymers-16-00714-t002:** Chemical composition of the “Leningradskoe” red crushed stone.

SiO_2_ %	TiO_2_ %	Al_2_O_3_ %	Fe_2_O_3_ %	CaO %	MgO %	S %	Mn ppm	K_2_O %	Na_2_O %
73.20	0.98	13.87	2.29	1.14	0.39	-	1051	4.63	3.50

**Table 3 polymers-16-00714-t003:** Roughness and hardness of mineral substrates.

Substrate	Roughness, *R_a_*, µm	Hardness, Rockwell Scale, HRC
Gray crushed stone	4.3	69.2
Red crushed stone	4.3	76.8

**Table 4 polymers-16-00714-t004:** Numerical values of levels for each factor.

Factors	Level 1	Level 2	Level 3	Level 4	Level 5
C_AS-1_, g/dm^3^ (x_1_)	0	0.5	1.0	1.5	2.0
C_AG-4I_, g/dm^3^ (x_2_)	0	0.5	1.0	1.5	2.0

**Table 5 polymers-16-00714-t005:** General view of the multilevel design matrix of a two-factor experiment.

X_1_ Factor Levels	X_2_ Factor Levels				
	1	2	3	4	5
1	y_1_	y_6_	y_11_	y_16_	y_21_
2	y_2_	y_7_	y_12_	y_17_	y_22_
3	y_3_	y_8_	y_13_	y_18_	y_23_
4	y_4_	y_9_	y_14_	y_19_	y_24_
5	y_5_	y_10_	y_15_	y_20_	y_25_

**Table 6 polymers-16-00714-t006:** Sampling an experimental array.

X_1_ Factor Levels C_AS-1_, g/dm^3^	Sample	X_2_ Factor Levels C_AG-4I_, g/dm^3^	Sample
0	(y_1_ + y_6_ + y_11_ + y_16_ + y_21_)/5	0	(y_1_ + y_2_ + y_3_ + y_4_ + y_5_)/5
0.5	(y_2_ + y_7_ + y_12_ + y_17_ + y_22_)/5	0.5	(y_6_ + y_7_ + y_8_ + y_9_ + y_10_)/5
1.0	(y_3_ + y_8_ + y_13_ + y_18_ + y_23_)/5	1.0	(y_11_ + y_12_ + y_13_ + y_14_ + y_15_)/5
1.5	(y_4_ + y_9_ + y_14_ + y_19_ + y_24_)/5	1.5	(y_16_ + y_17_ + y_18_ + y_19_ + y_20_)/5
2.0	(y_5_ + y_10_ + y_15_ + y_20_ + y_25_)/5	2.0	(y_21_ + y_22_ + y_23_ + y_24_ + y_25_)/5

**Table 7 polymers-16-00714-t007:** Indicators of surface activity of modifiers.

AG-4I		AAMDOR-10		AS-1	
*g*, mN×dm^3^/m×g	*R^2^*	*g*, mN×dm^3^/m×g	*R^2^*	*g*, mN×dm^3^/m×g	*R* ^2^
4.08	0.99	2.04	0.99	5.16	0.99

**Table 8 polymers-16-00714-t008:** Surface tension of ternary systems at the boundary with air.

Additive, g/dm^3^						
AS-1	AG-4I	Δ*σ_AS-_*_1_	Δ*σ_AG-_*_4*I*_	*σ_c_*, mN/m	*σ_ex_*, mN/m	Δ, mN/m
0.5	0.5	2.58	2.04	40.77	41.20	0.43
1.0	0.5	4.59	2.04	38.76	38.20	−0.56
1.5	0.5	4.49	2.04	38.86	38.75	−0.11
2.0	0.5	4.39	2.04	38.96	39.40	0.44
0.5	1.0	2.58	4.08	38.73	38.50	−0.23
1.0	1.0	4.59	4.08	36.72	37.20	0.48
1.5	1.0	4.49	4.08	36.82	37.00	0.18
2.0	1.0	4.39	4.08	36.92	37.30	0.38
0.5	1.5	2.58	3.09	39.72	39.20	−0.52
1.0	1.5	4.59	3.09	37.71	37.50	−0.21
1.5	1.5	4.49	3.09	37.81	37.80	−0.01
2.0	1.5	4.39	3.09	37.91	38.20	0.29
0.5	2.0	2.58	−0.17	42.98	42.50	−0.48
1.0	2.0	4.59	−0.17	40.97	40.50	−0.47
1.5	2	4.49	−0.17	41.07	40.70	−0.37
2	2	4.39	−0.17	41.17	40.90	−0.27

**Table 9 polymers-16-00714-t009:** The contact angle of wetting the surface of gray and red crushed stones with binary systems “bitumen–additive”.

C, g/dm^3^	*θ*, °			*θ*, °		
	Gray Crushed Stone			Red Crushed Stone		
	AG-4I	AS-1	AMDOR-10	AG-4I	AS-1	AMDOR-10
0	125.66	125.66	125.66	126.87	126.87	126.87
0.5	116.22	117.81	119.83	122.20	120.21	123.63
1.0	112.50	115.10	116.80	118.30	116.30	122.60
1.5	113.11	115.63	117.32	118.82	118.22	123.94
2.0	114.32	117.84	119.35	119.71	120.44	124.41

**Table 10 polymers-16-00714-t010:** Wetting activity of modifying additives on surfaces of various types.

Additive	AG-4I		AMDOR-10		AS-1	
The nature of crushed stone	*a*, dm^3^/g	*R* ^2^	*a*, dm^3^/g	*R* ^2^	*a*, dm^3^/g	*R* ^2^
Gray crushed stone	0.19	0.93	0.12	0.95	0.15	0.91
Red crushed stone	0.13	0.99	0.09	0.99	0.16	0.98

**Table 11 polymers-16-00714-t011:** Relative changes in the specific surface energy of the solid surface of crushed stone.

	Gray Crushed Stone						Red Crushed Stone					
	AG-4I		AS-1		AMDOR-10		AG-4I		AS-1		AMDOR-10	
C, g/dm^3^	Δ*σ_s_*, mN/m	α, %	Δ*σ_s_*, mN/m	α, %	Δ*σ_s_*, mN/m	α, %	Δ*σ_s_*, mN/m	α, %	Δ*σ_s_*, mN/m	α, %	Δ*σ_s_*, mN/m	α, %
0	−26.33		−26.33		−26.33		−27.23		−27.23		−27.23	
0.5	−19.07	27.57	−20.12	23.59	−22.19	15.72	−22.98	15.61	−21.41	21.37	−24.40	10.39
1.0	−15.70	40.37	−17.14	34.90	−20.12	23.59	−19.42	28.68	−17.95	34.08	−24.14	11.35
1.5	−16.69	36.61	−17.59	33.19	−20.65	21.57	−20.89	23.28	−19.23	29.38	−25.14	7.68
2.0	−18.68	29.05	−19.27	26.81	−22.16	15.84	−22.78	16.34	−20.91	23.21	−25.50	6.35

**Table 12 polymers-16-00714-t012:** The values of the contact angle of the surface of crushed stone with ternary systems “bitumen–AG-4I–AS-1”.

**Gray Crushed Stone**									
**C_AG-4I_, g/dm^3^**	** *θ* ** **, °**	**C_AS-1_,** **g/dm^3^**	** *θ* ** **, °**	**C_AS-1_,** **g/dm^3^**	** *θ* ** **, °**	**C_AS-1_,** **g/dm^3^**	** *θ* ** **, °**	**C_AS-1_,** **g/dm^3^**	** *θ* ** **, °**
0	125.66	0.5	117.83	1.0	115.14	1.5	115.62	2.0	117.82
0.5	116.23	0.5	113.81	1.0	112.60	1.5	112.82	2.0	112.93
1.0	113.10	0.5	111.0	1.0	109.70	1.5	111.61	2.0	112.34
1.5	112.53	0.5	111.22	1.0	110.82	1.5	110.54	2.0	112.30
2.0	114.30	0.5	113.91	1.0	111.52	1.5	112.43	2.0	113.20
**Red Crushed Stone**									
**C_AG-4I_, g/dm^3^**	** *θ* ** **, °**	**C_AS-1_,** **g/dm^3^**	***θ*, °**	**C_AS-1_,** **g/dm^3^**	** *θ* ** **, °**	**C_AS-1_,** **g/dm^3^**	** *θ* ** **, °**	**C_AS-1_,** **g/dm^3^**	** *θ* ** **, °**
0	126.87	0.5	120.23	1.0	116.33	1.5	118.20	2.0	120.42
0.5	122.22	0.5	120.15	1.0	118.52	1.5	118.92	2.0	119.24
1.0	118.35	0.5	115.61	1.0	112.81	1.5	113.21	2.0	114.43
1.5	118.81	0.5	117.22	1.0	115.14	1.5	114.51	2.0	115.65
2.0	119.72	0.5	118.84	1.0	116.51	1.5	117.30	2.0	117.82

**Table 13 polymers-16-00714-t013:** Reducing the specific surface energy of a solid surface.

	**Gray Crushed Stone**							
	**C_AG-4I_ = 0.5 g/dm^3^**		**C_AG-4I_ = 1.0 g/dm^3^**		**C_AG-4I_ = 1.5 g/dm^3^**		**C_AG-4I_ = 2.0 g/dm^3^**	
**C_AS-1_, g/dm^3^**	**Δ*σ*_s_,** **mN/m**	**α, %**	**Δ*σ_s_*, mN/m**	**α, %**	**Δ*σ_s_*, mN/m**	**α, %**	**Δ*σ_s_*, mN/m**	**α, %**
0	−26.33		−26.33		−26.33		−26.33	
0.5	−16.63	36.86	−13.92	47.12	−14.05	46.65	−17.22	34.61
1.0	−14.68	44.25	−12.54	52.37	−13.32	49.42	−14.84	43.63
1.05	−15.02	42.97	−13.62	48.27	−13.24	49.72	−15.51	41.10
2.0	−15.33	41.77	−14.15	46.25	−14.50	44.95	−16.11	38.81
	**Red Crushed Stone**							
	**C_AG-4I_ = 0.5 g/dm^3^**		**C_AG-4I_ = 1.0 g/dm^3^**		**C_AG-4I_ = 1.5 g/dm^3^**		**C_AG-4I_ = 2.0 g/dm^3^**	
**C_AS-1_,** **g/dm^3^**	**Δ*σ_s_*,** **mN/m**	**α, %**	**Δ*σ_s_*,** **mN/m**	**α, %**	**Δ*σ_s_*, mN/m**	**α, %**	**Δ*σ_s_*, mN/m**	**α, %**
0	−26.33		−26.33		−26.33		−26.33	
0.5	−20.66	24.12	−16.64	38.91	−17.92	34.20	−20.47	24.81
1.0	−18.23	33.06	−14.42	47.06	−15.91	41.58	−18.07	33.64
1.5	−18.73	31.23	−14.58	46.47	−15.68	42.43	−18.67	31.45
2.0	−19.22	29.41	−15.41	43.41	−16.51	39.38	−19.08	29.95

**Table 14 polymers-16-00714-t014:** Adhesive efficiency of modifying additives in the binary compositions in relation to the surface of crushed stone.

C_AG-4I_	*A*, %	C_AS-1_	*A*, %	C_AMDOR-10_	*A*, %
Gray crushed stone					
0.5	30.52	0.5	28.02	0.5	24.15
1.0	38.67	1.0	29.32	1.0	25.06
1.5	34.92	1.5	28.51	1.5	24.92
2.0	32.88	2.0	26.90	2.0	23.33
Red crushed stone					
0.5	20.98	0.5	26.55	0.5	20.82
1.0	24.25	1.0	28.76	1.0	21.06
1.5	22.67	1.5	24.21	1.5	20.32
2.0	20.56	2.0	23.02	2.0	20.00

**Table 15 polymers-16-00714-t015:** The adhesive efficiency of modifying additives in the ternary compositions in relation to the surface of crushed stone.

**Gray Crushed Stone**									
**C_AG-4I_, g/dm^3^**	***A*, %**	**C_AS-1_,** **g/dm^3^**	***A*, %**	**C_AS-1_,** **g/dm^3^**	***A*, %**	**C_AS-1_,** **g/dm^3^**	***A*, %**	**C_AS-1_,** **g/dm^3^**	***A*, %**
0	0	0.5	28.02	1.0	29.32	1.5	28.51	2.0	26.90
0.5	30.52	0.5	36.52	1.0	43.43	1.5	42.01	2.0	43.34
1.0	38.67	0.5	44.89	1.0	52.28	1.5	43.47	2.0	39.26
1.5	34.92	0.5	47.87	1.0	43.76	1.5	43.15	2.0	39.82
2.0	32.88	0.5	41.03	1.0	48.75	1.5	45.78	2.0	37.91
**Red Crushed Stone**									
**C_AG-4I_, g/dm^3^**	***A*, %**	**C_AS-1_,** **g/dm^3^**	***A*, %**	**C_AS-1_,** **g/dm^3^**	***A*, %**	**C_AS-1_,** **g/dm^3^**	***A*, %**	**C_AS-1_,** **g/dm^3^**	***A*, %**
0	0	0.5	26.55	1.0	28.76	1.5	24.21	2.0	23.02
0.5	20.98	0.5	28.70	1.0	32.67	1.5	31.00	2.0	31.20
1.0	24.25	0.5	35.98	1.0	36.70	1.5	30.48	2.0	30.81
1.5	22.67	0.5	35.05	1.0	36.15	1.5	37.18	2.0	35.76
2.0	20.56	0.5	30.15	1.0	31.03	1.5	30.00	2.0	29.25

**Table 16 polymers-16-00714-t016:** Multilevel plan-matrix of a two-factor experiment.

C_AS-1_, g/dm^3^	C_AG-4I_, g/dm^3^				
	0	0.5	1.0	1.5	2.0
Gray Crushed Stone					
0.0	−0.583	−0.442	−0.392	−0.383	−0.412
0.5	−0.466	−0.404	−0.362	−0.358	−0.405
1.0	−0.424	−0.384	−0.337	−0.355	−0.367
1.5	−0.432	−0.388	−0.368	−0.350	−0.381
2.0	−0.466	−0.389	−0.379	−0.379	−0.394
Red Crushed Stone					
0.0	−0.600	−0.533	−0.474	−0.482	−0.495
0.5	−0.503	−0.502	−0.432	−0.457	−0.482
1.0	−0.443	−0.477	−0.388	−0.424	−0.446
1.5	−0.473	−0.483	−0.394	−0.415	−0.459
2.0	−0.506	−0.488	−0.413	−0.432	−0.466

**Table 17 polymers-16-00714-t017:** Sample of response functions for each level of concentration factors of two additives.

**C_AS-1_, g/dm^3^**	**Gray Crushed Stone**	**Red Crushed Stone**
0.0	−0.442	−0.517
0.5	−0.399	−0.475
1.0	−0.373	−0.436
1.5	−0.384	−0.445
2.0	−0.402	−0.461
**C_AG-4I_, g/dm^3^**	**Gray Crushed Stone**	**Red Crushed Stone**
0.0	−0.474	−0.505
0.5	−0.401	−0.497
1.0	−0.368	−0.420
1.5	−0.365	−0.442
2.0	−0.392	−0.470

## Data Availability

The datasets generated and/or analyzed during the current study are available from the corresponding author upon reasonable request.

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
