# Peer review of "Increasing the Adhesion of Bitumen to the Surface of Mineral Fillers through Modification with a Recycled Polymer and Surfactant Obtained from Oil Refining Waste"

_polymers, 2024, doi:10.3390/polym16050714_

Round 1

Reviewer 1 Report

Comments and Suggestions for Authors

The manuscript introduces the process of changing the additive content in asphalt mineral components to achieve better mixture performance. The overall paragraphs of the manuscript are clear, the grammar is correct, and it meets the publication requirements of the journal. The following are the specific opinions:

1. Verify and maintain consistency in the format of the charts.

2. There is too much content in the conclusion section that needs to be simplified, highlighting the conclusive content.

Author Response

  1. We have corrected the charts format.
  2. We have shortened the conclusion section.

Reviewer 2 Report

Comments and Suggestions for Authors

The authors presented the study on "Increasing the Adhesion of Bitumen to the Surface of the Min- 2 eral Fillers by Modification with a Recycled Polymer and Sur- 3 factant Obtained from Oil Refining Waste". They presented the study perfectly why the study is important. 

The methods section was well done. I think adding a bit more detail on some of the techniques used could be helpful for readers who might not be super familiar with them.

The discussions section also well written.

Main Question Addressed: The research investigates the optimization of bitumen-mineral compositions by modifying them with specific additives (surfactant and polymer) to enhance adhesion to crushed stone surfaces, assessed through the wetting contact angle criterion.

 Originality and Relevance: The research is relevant as it explores the enhancement of bitumen adhesion, a critical aspect in construction and infrastructure. The novelty of the current research is incorporation of recycled polymer and surfactant from oil refining waste to modify bitumen, addressing a gap in sustainable additives for bitumen.

 Contribution to the Field: It adds value by presenting a novel approach using recycled additives to enhance bitumen's adhesion to mineral surfaces. This approach differs from conventional methods and contributes to sustainable practices in construction materials.

 Methodology Improvements: While the study provides valuable insights, further details on the methodology could enhance its robustness. Specific improvements might involve expanding the experimental controls, exploring a wider range of concentrations for additives, and possibly examining the long-term durability of the modified compositions.

 Consistency of Conclusions: The conclusions align with the presented evidence, demonstrating how the joint presence of specific additives optimizes adhesion between bitumen and crushed stone. The study's findings regarding the wetting activity of additives and their impact on surface energy are well-supported.

 Appropriateness of References: The references used seem relevant and support the study's context in understanding the properties of additives and their impact on bitumen compositions.

 Additional Comments on Tables and Figures: All the tables and figures provided by the authors were sufficient to the research and found to be good quality.

 Given the substantial contributions and the relevance of the study in addressing sustainability in construction materials, I recommend accepting the paper for publication. However, the authors should address the suggestions for methodological improvements and ensure clarity in data representation for a comprehensive understanding by the readers.

Author Response

We have added a little more detailed information about some of the methods (lines 199, 206, 211, 213, 215.

Reviewer 3 Report

Comments and Suggestions for Authors

Author Response

  1. We have corrected the IR spectrums.
  2. We have removed Figures 7, 9, 10.
  3. We have added the information about wettability and static contact angle (lines 251-258).
  4. Figure 11 b - we have added the caption (Figure 8 b).
  5. We used the "individual measurements" instead of "parallel measurements".
  6. We have used "gray-crushed stone" and "red-crushed stone".
  7. Figures 14-15 (now 11-12) - we have deleted "of wetting" on y-axis.
  8. We have added roughness parameters for the surface PMMA and PVC (lines 566, 584).

Round 2

Reviewer 3 Report

Comments and Suggestions for Authors

The authors have addressed my comment. This manuscript can be accepted in present from.